# Effects of Auxin (Indole-3-butyric Acid) on Adventitious Root Formation in Peach-Based *Prunus* Rootstocks

**DOI:** 10.3390/plants11070913

**Published:** 2022-03-29

**Authors:** María Salud Justamante, Mariem Mhimdi, Marta Molina-Pérez, Alfonso Albacete, María Ángeles Moreno, Inés Mataix, José Manuel Pérez-Pérez

**Affiliations:** 1Instituto de Bioingeniería, Universidad Miguel Hernández, 03202 Elche, Spain; mjustamante@umh.es (M.S.J.); mmhimdi@umh.es (M.M.); marta.molina03@alu.umh.es (M.M.-P.); 2Departmento de Nutrición Vegetal, CEBAS-CSIC, 30100 Murcia, Spain; alfonsoa.albacete@carm.es; 3Department of Pomology, Estación Experimental de Aula Dei-CSIC, 50059 Zaragoza, Spain; mmoreno@eead.csic.es; 4Invisa Biotecnología Vegetal S.L., 30410 Caravaca de la Cruz, Spain; imataix@nurfruits.com

**Keywords:** vegetative propagation, *Prunus* rootstocks, hormone profiling, auxin homeostasis

## Abstract

Several *Prunus* species are among the most important cultivated stone fruits in the Mediterranean region, and there is an urgent need to obtain rootstocks with specific adaptations to challenging environmental conditions. The development of adventitious roots (ARs) is an evolutionary mechanism of high relevance for stress tolerance, which has led to the development of environmentally resilient plants. As a first step towards understanding the genetic determinants involved in AR formation in *Prunus* sp., we evaluated the rooting of hardwood cuttings from five *Prunus* rootstocks (Adafuel, Adarcias, Cadaman, Garnem, and GF 677) grown in hydroponics. We found that auxin-induced callus and rooting responses were strongly genotype-dependent. To investigate the molecular mechanisms involved in these differential responses, we performed a time-series study of AR formation in two rootstocks with contrasting rooting performance, Garnem and GF 677, by culturing in vitro microcuttings with and without auxin treatment (0.9 mg/L of indole-3-butyric acid [IBA]). Despite showing a similar histological structure, Garnem and GF677 rootstocks displayed dynamic changes in endogenous hormone homeostasis involving metabolites such as indole-3-acetic acid (IAA) conjugated to aspartic acid (IAA-Asp), and these changes could explain the differences observed during rooting.

## 1. Introduction

The root system of plant species is of great importance for plant anchorage and the absorption of water and nutrients from the soil [1]. Adventitious rooting is a multifactorial response that leads to the formation of new roots from aerial organs and, if necessary, to the establishment of a complete and autonomous plant when the main root system is absent [2]. Wound-induced adventitious root (AR) formation is crucial for the clonal propagation of forest and horticultural species. Deprivation of the original root system interrupts the supply of water, nutrients, and plant hormones, such as cytokinins formed in the roots, which in turn leads to the accumulation of other downwardly transported metabolites, such as auxin, at the basal region of the stem near the wound [3]. In response to excision, a new developmental program is initiated at the base of the stem near the wound, ultimately leading to the generation of a new root system.

The *Prunus* genus belongs to the Rosaceae family and includes approximately 200 species, some of which are commercially important, such as the cherry tree (*Prunus cerasus* L.), the peach tree (*Prunus persica* (L.) Batsch), or the almond tree (*Prunus dulcis* (Mill.) DA Webb, syn. *Prunus amygdalus* Batsch) [4]. The economic importance of this genus of stone fruit trees lies in the various uses of its species as a source of food and other resources such as wood or oil, in addition to its use as ornamental plants [4]. The commercial production of many species of the genus *Prunus* involves the use of appropriate rootstocks belonging to the same or other species of the same genus. The rootstocks constitute the main support of the shoots of the varieties of interest, being responsible for the absorption of water and nutrients, and also for providing resistance to soil pathogens, as well as tolerance to stressful environmental conditions [5]. In Spain, the economic importance of some *Prunus* species is unquestionable, placing this country as the first European producer of peach and nectarine with more than one million tons [6,7]. However, the cultivation of these plant species entails a series of shortcomings and problems that must be addressed to ensure the sustainability and economic profitability of the crops [6]. The presence of limestone soils in the Mediterranean region implies abiotic stresses that limit the cultivation of most stone fruit species. In these soils, iron chlorosis and root asphyxia predominate, both associated with tree mortality and nutrient deficiency [8,9,10]. In addition, the increasing salinization of soils and the scarce availability of water are other relevant problems that also acquire special relevance in the Mediterranean region [8]. Likewise, the climate change experienced in the last two decades had serious effects, highlighting an increase in summer temperatures and a shorter duration of the winter period, which results in the advancement of the flowering date and the consequent increase in the risk against spring frost [6]. Thus, the search for varieties with lower winter cold requirements and with flowering appropriate to the climatic zone of their cultivation are priority objectives in rootstock improvement programs in most European countries [8,11].

GF 677 is known to be the most widely used peach–almond hybrid in the Mediterranean region and shows greater adaptability to limestone and low nutrient soils, which are common in this region [10,11]. On the contrary, despite the high vigor of the Cadaman rootstock, it is known as a hard-to-root genotype [12]. Garnem is widely used as a rootstock due to its high adaptability in both irrigated and non-irrigated soils [13], although waterlogging conditions should be avoided [10,14]. Furthermore, previous results show that Garnem is easily propagated both as hardwood cuttings and in vitro grown microshoot cuttings [9]. In addition, other rootstocks such as Adafuel and Adarcias display high to moderate rooting capacity, respectively, as previously described [15], and are suitable for peach production in low nutrient calcareous soils [14,16].

Only two complete reference genomes are described in *Prunus* sp., the peach genome and the Japanese apricot genome [17]. In this sense, the use of new high-throughput sequencing methods will allow us to improve our understanding of the molecular networks that regulate the biology of this species, information that can be used in breeding programs [17]. In addition, the development of new, more effective auxins, as well as the identification of possible rooting cofactors (auxin synergists), are also critical areas of research, which will be key in the vegetative propagation of elite genotypes and the expansion of stone fruit species cultivation [18].

In this work, we studied the rooting performance of five *Prunus* rootstocks, in which the differences in rooting capacity under substrate conditions were confirmed. Since the study of rooting capacity under field conditions is difficult to carry out, we established an experimental design based on hydroponic culture, which allowed us to quantify the differences in regeneration among the studied genotypes. Based on the results obtained from this study, two genotypes with contrasting rooting behavior were selected for detailed in vitro phenotypic and hormonal analyses.

## 2. Results

### 2.1. Rooting Response in Five Prunus Rootstocks during Hydroponic Culture

This project was initiated to evaluate the rooting performance of several *Prunus* rootstocks used for the mass production of grafted stone fruit trees: four peach–almond hybrids, Adafuel, Adarcias, Garnem, and GF 677, and one peach–Chinese wild peach hybrid, Cadaman (Table 1). The selected rootstocks represent a wide range of genetic backgrounds and showed, in previous studies, large differences in some traits related to rooting [11,16,19]. GF 677 and Cadaman were chosen due to their great use in Spain, since both rootstocks represented 71% of the total number of rootstocks used, with 50% and 21%, respectively [11].

By conducting a small-scale trial, we confirmed that the rooting response of hardwood stem cuttings after auxin treatment and grown for 82 days in a commercial substrate under controlled environmental conditions was highly genotype-dependent (Appendix A). For a detailed characterization of the contrasting rooting responses observed in hardwood cuttings from selected genotypes, we devised a hydroponic culture system. This system allowed us to periodically evaluate multiple parameters associated with effective rooting (Appendix A), such as the percentage of callus response or the number of ARs, which were measured at 32, 50, and 90 days after planting (dap). AR performance was also estimated by establishing different categories according to the number of ARs formed at 90 dap (Appendix A), and a visual callus-stage scale was also established (Appendix A).

#### 2.1.1. Callus Formation and Rooting Efficiency

Adafuel and Garnem were the first genotypes to initiate a regenerative response in the basal region of the stem, with 83.5% and 46.3% callus formation at 32 dap, respectively (Figure 1a). The earliest formation of ARs was observed in Garnem, with 6.0% of cuttings rooted at 32 dap (Figure 1a). At 90 dap, Adafuel reached the highest regenerative response, followed by Adarcias, Garnem, and GF 677 (Figure 1b). Adafuel and Garnem presented the highest percentage of rooted cuttings, with 70.0% and 67.4%, respectively. In contrast, Cadaman had the lowest regenerative response in terms of the callus and AR formation of the studied genotypes, and the percentage of rooted cuttings only reached 3.4% at 90 dap (Figure 1a,b).

In addition to the regenerative response (which was highest in Adafuel), the effective rooting of hardwood cuttings was dependent on the development of many functional ARs. Garnem and GF 677 cuttings were the first genotypes to develop 10 or more ARs at 50 dap (Figure 1c). At the end of the experiment (Figure 1d), the rooted Garnem cuttings had the highest number of ARs (8.8 ± 1.2), followed by GF 677 (7.0 ± 1.8) and Adafuel (6.1 ± 0.5). Therefore, the adventitious rooting of hardwood cuttings was effectively evaluated under hydroponic conditions (Figure 1e and Appendix A), as both the regenerative responses and rooting capacity of the cuttings confirmed the contrasting characteristics of the studied genotypes under commercial conditions.

#### 2.1.2. Correlations between Regenerative Responses

We noticed that, in all cases, callus formation in the basal region of the cuttings preceded AR initiation. We measured the callus area at the end of the experiment from stored images, and the highest values were found in Adafuel (0.82 ± 0.05 cm^2^), followed by Garnem (0.56 ± 0.06 cm^2^), Adarcias (0.41 ± 0.03 cm^2^), and Cadaman (0.21 ± 0.07 cm^2^). GF 677 showed the lowest average callus area (0.19 ± 0.02 cm^2^). For the rapid determination of callus growth, we established a visual classification of callus stages (Appendix A). Callus stage was positively correlated with the measured callus area (*p*-value = 0.000, Pearson) (Appendix A), confirming that the visual assignment of callus stage is a reliable parameter to qualitatively assess callus growth. Furthermore, we found no significant association (*p*-value > 0.05; Pearson) between the callus area, AR number, and maximum AR length (Appendix A), suggesting the independent regulation of these regenerative responses.

Due to the low rooting performance of Adarcias and Cadaman, these two rootstocks were discarded for further studies. Of the other three rootstocks, Garnem and GF 677 were selected to be studied during in vitro culture since both showed significant differences in the percentage of AR response (*p*-value < 0.05; Chi-square) as well as in the AR number (*p*-value < 0.01).

### 2.2. Regenerative Response of Garnem and GF 677 during In Vitro Rooting

We performed two in vitro experiments (E1 and E2) with Garnem and GF 677 microcuttings provided by a private company (Appendix A, see Section 4). They remained in the preincubation medium for 20 and 11 days, respectively. The imaging of explants at different time points (Appendix A) allowed us to quantify different regeneration parameters. We measured the AR response and rooting capacity of the starting material (Appendix A) and found that in both cases, Garnem showed higher rooting performance than GF 677 (Appendix A). Furthermore, the rooting performance of the starting material was positively correlated with the time spent in the preincubation medium (Appendix A).

#### 2.2.1. Histology

To understand the cellular events leading to the contrasting AR responses observed within the studied genotypes, we analyzed cross-sections of the basal region of the microcuttings. Some structural elements could be identified in these sections: pith, xylem, cambium, phloem, cortex, and epidermis (Figure 2a,d). The growth of the stem microcuttings in the preincubation medium caused the stems to initiate secondary growth. In fact, we observed regular organization in the cells of the epidermis, cortex, and pith, while the cambium cells were highly vacuolated and with defined nuclei. No striking anatomical differences were found between the two studied genotypes.

#### 2.2.2. Auxin Treatment Improves Callus Formation and Enhances AR Number

After we pooled all the data from E1 and E2, we performed a multivariate ANOVA to determine which factor(s) contributed the most to the observed variation (Appendix A). No significant differences (*p*-value > 0.05) were found in the AR number at 15 days after excision (dae) with respect to the experiment (Appendix A), container, or spatial position within the container (Appendix A). These results indicate that, despite the differences in starting material, the timing of wounding activates the regeneration response similarly, so this point was defined as 0 dae.

The regenerative response was periodically analyzed based on the percentage of callus and AR formation after the excision of the basal region of the microcuttings (Figure 3). Without exogenous auxin, both rootstocks were able to develop calluses at the base of the stem all through the wound, but callus emergence in Garnem arose earlier than in GF 677 (Figure 3a). Treatment with 0.9 mg/L of indole-3-butyric acid (IBA) had a significant effect in anticipating callus formation in Garnem and GF 677 (Figure 3a). A similar trend was observed for AR formation (Figure 3b) but with some delays in callus formation. In all cases, callus formation preceded AR formation, and the IBA-treatment anticipated AR formation in Garnem and GF 677. For each genotype and treatment, we estimated the time when half of the explants showed a regenerative response (i.e., producing callus or ARs), which we dubbed as the RR50 (Table 2).

We found significant differences (*p*-value < 0.05; Chi-square) in callus and AR formation in terms of genotype and treatment (Figure 3c,d). At 9 dae, the highest percentage of callus formation was found in Garnem treated with IBA (81.3%), where about 60% of their explants developed ARs (Figure 3c). In both genotypes, the IBA-supplemented medium produced a significant increase (*p*-value < 0.05; Chi-square) in callus and AR formation, although GF 677 showed lower values for callus and AR formation than Garnem (Appendix A). Interestingly, both IBA-treated GF 677 explants and non-treated Garnem explants displayed non-significant differences (*p*-value = 0.07; Chi-square) in callus and AR formation (Figure 3d), confirming that the addition of exogenous auxin can improve the poor rooting response of GF 677 microcuttings. IBA-supplemented medium also significantly (*p*-value = 0.0000; Chi-square) improved the regenerative response in Garnem, as almost all explants developed ARs at the end of the experiment (Figure 3d).

The results corresponding to the rooting capacity at 30 dae are shown in Figure 3e. IBA-treated Garnem microcuttings developed eight or more roots per explant in more than half of them (65.7%). In this way, GF 677 treated with IBA reached a similar AR capacity than non-treated Garnem, while the number of ARs per microcutting increased significantly (*p*-value < 0.01) in Garnem treated with IBA (Figure 3e). Although the number of ARs in IBA-treated microcuttings increased relative to those grown in mock medium, the length of the ARs in the IBA-supplemented medium was much shorter (Figure 3f).

#### 2.2.3. Improved Rooting Performance on *Prunus* Rootstocks by an IBA Pulse

As it is well known that prolonged incubation with auxin can cause excessive callus formation, which could limit effective rooting [20], we tested a 24 h incubation treatment with 0.9 mg/L IBA in a third experiment (E3), hereinafter referred to as the IBA pulse (IBAp). We found that IBAp-treated microcuttings showed premature callus formation compared to microcuttings grown on medium without IBA (i.e., mock condition) (Appendix A). In GF 677 in mock condition, callus formation began around 8 dae, while IBAp treatment induced earlier callus formation in GF 677, at a rate similar to untreated Garnem (Figure 4a). The IBAp treatment of Garnem also improved callus formation (Figure 4a), but to a similar degree as continuous IBA treatment (Figure 3a). Interestingly, IBAp treatment markedly decreased the initiation of AR formation in both genotypes (Figure 4b). In fact, AR formation in IBAp-treated microcuttings was observed at 7 dae in 38.1% and 16.7% of Garnem and GF 677, respectively (Appendix A). Unlike the treatment with continuous IBA (Figure 3), the treatment with IBAp produced AR initiation in GF 677 even earlier than Garnem in mock conditions and, subsequently, the percentage of AR formation at 9 dae was significantly higher (*p*-value = 0.0000; Chi-square) in GF 677 IBAp-treated than in untreated Garnem (Figure 4c). However, at 20 dae there were no significant differences (*p*-value = 0.088; Chi-square) between Garnem in mock conditions and IBAp-treated GF 677 regarding the regenerative response of the microcuttings (Figure 4d). Therefore, IBAp treatment accelerated AR initiation by 6.0 days in Garmen and 11.9 days in GF 677, arising in both cases from previously established calluses (Table 3 and Appendix A).

Rooting capacity at the end of the experiment (20 dae) was significantly improved (*p*-value < 0.01) by IBAp in both genotypes (Figure 4e). About 50–60% of GF 677 and Garnem microcuttings produced more than five ARs on IBAp, and around 20% of Garmen microcuttings on IBAp had more than eight ARs (Figure 4e). Unlike what was found in continuous IBA incubation where ARs were short (Figure 3f), IBAp-treated ARs were able to elongate to a similar degree as untreated microcuttings (Figure 4f).

We then compared the rooting performance of the different treatments at 20 dae by studying the number and the maximum length of the ARs (Appendix A). These results confirmed that IBAp had a positive effect on effective rooting in both genotypes. On the one hand, GF 677 IBAp displayed a significant increase (*p*-value < 0.01) both in the number of ARs (3.6 ± 0.6) and in the maximum AR length (66.7 ± 3.7 mm) compared to those found in mock condition (0.8 ± 0.4 ARs and 24.2 ± 7.3 mm) (Appendix A). On the other hand, although Garnem produced more ARs in IBA (8.1 ± 0.45 ARs) than in IBAp (5.7 ± 0.7 ARs), their ARs were longer in IBAp conditions (62.5 ± 5.4 mm) (Appendix A). In fact, IBAp treatment produced higher shoot growth of both genotypes compared to untreated or continuous IBA conditions (Appendix A), confirming the better performance of the AR system.

We wondered if IBAp could trigger AR formation after a short incubation under in vitro conditions. In a further trial, microcuttings were maintained for 96 h under in vitro conditions after IBAp treatment and then transferred to substrate conditions for about three weeks. We found significant differences (*p*-value < 0.05, Chi-square) in plant survival and shoot growth due to IBAp treatment (Appendix A). At the end of the experiment, the number of ARs was significantly higher (*p*-value < 0.05) in Garnem and GF 677 microcuttings that were treated with IBAp, compared to the untreated ones (Appendix A). Furthermore, the AR system was highly developed in IBAp-treated microcuttings (Appendix A). Taken together, these results indicate that a short pulse of IBA is sufficient to trigger AR initiation in *Prunus* microcuttings, shortening the time required under in vitro conditions.

#### 2.2.4. Hormone Profiling of Garnem and GF 677 Microcuttings during IBA-Induced Rooting

Next, we measured the endogenous levels of 17 hormone metabolites in the basal region of microcuttings at 0, 4, and 8 dae (see Section 4). For most of them, we found non-significant differences (*p*-value > 0.05) in their endogenous levels regarding time and replicates (Appendix A). Indole-3-acetic acid (IAA) levels were similar in Garnem and GF 677 at excision time (0 dae) and significantly increased (*p*-value = 0.000) with IBA treatment in both genotypes and at similar levels (Figure 5a). In addition, the levels of IAA conjugated with isoleucine (IAA-Ile) and with aspartic acid (IAA-Asp) were slightly increased in GF 677 with respect to those found in Garnem at 0 dae (Figure 5b and Appendix A). In all cases, IBA treatments significantly (*p*-value < 0.005) increased the endogenous levels of inactive IAA derivatives, such as IAA-Ile or IAA-Asp, as well as those of methyl-IAA (MeIAA) (Figure 5b and Appendix A). Several bioactive cytokinins were measured: *trans*-zeatin (*t*Z), *cis*-zeatin (*c*Z), dihydrozeatin (DHZ), and N6-(Δ2-Isopentenyl)adenine (iP) (Appendix A). The levels of *t*Z, *c*Z, and DHZ in the basal region of the microcuttings were not significantly different (*p*-value > 0.05) between Garnem and GF 677 at the time of excision (0 dae), which were otherwise significantly (*p*-value < 0.01) increased by IBA treatment, either by continuous or pulsed IBA incubation (Figure 5c and Appendix A). These results indicate that exogenous IBA mainly induced the accumulation of *t*Z and *c*Z in the basal region of the microcuttings. On the other hand, iP levels were low and mostly unchanged in the experimental dataset (Appendix A). Gibberellic acid (GA3) and gibberellin A4 (GA4) were detected (Appendix A), but only GA3 gave consistent results and was analyzed further. Interestingly, GA3 levels were approximately three-fold lower (*p*-value < 0.001) in GF 677 at 0 dae than in Garnem, and their levels were reduced by IBA treatments in this later genotype (Figure 5d). Although we detected active strigolactones, epibrassinolide (EpiBL) and solanacol (SL) in the basal region of Garnem and GF 677 microcuttings, their levels did not change significantly (*p*-value > 0.05) during the experiment and were not studied further (Appendix A).

A striking pattern of 1-aminocyclopropane 1-carboxylic acid (ACC) accumulation was found in the microcuttings of GF 677 and Garnem during IBA-induced rooting (Appendix A). ACC is an ethylene precursor whose levels were previously used as an indirect estimate of ethylene concentration [21]. When Garnem was grown in a medium without IBA, ACC levels were highest at 0 dae and decreased during rooting. Furthermore, ACC levels in GF 677 increased during rooting both in mock conditions and in IBA-treated microcuttings (Figure 5e). A similar pattern was also found for abscisic acid (ABA), with the highest levels of this hormone found in GF 677 after IBA treatments (Figure 5f). ABA levels remained stable in Garnem at different times between treatments but decreased significantly in mock conditions. Endogenous levels of jasmonic acid (JA) and methyl jasmonate (MeJA) were dependent on time after excision (*p*-value = 0.01) and treatment (*p*-value = 0.0001) (Appendix A). Notably, JA and MeJA levels increased after excision and during rooting after IBA treatments and at higher levels in GF 677 than in Garnem (Figure 5g and Appendix A). For salicylic acid (SA), GF 677 contained significantly higher levels of SA than Garnem, and these were not significantly changed (*p*-value > 0.001) by IBA treatments in both genotypes (Figure 5h). In general, we found slightly reduced levels of some stress hormones at excision time, such as ACC (ethylene precursor), ABA, and SA, in GF 677 compared to those found in Garnem, but their levels were significantly enhanced by all of the IBA treatments, mainly in GF 677. In contrast, GF 677 had higher IAA-Asp levels than Garnem at excision time (Appendix A), which could explain its reduced rooting in mock conditions. We found no striking differences in hormonal profiles regarding continuous IBA or IBA-pulse treatments, suggesting that a short IBA treatment could regulate hormone homeostasis during subsequent rooting.

## 3. Discussion

Despite the availability of effective protocols for vegetative multiplication in most woody plants, the rooting efficiency of some *Prunus* sp. clones is problematic [22]. Rootstocks are an essential component of modern fruit agriculture due to their ability to adapt a particular crop to various environmental conditions and cultural practices [23]. The use of some *Prunus* sp. as rootstocks that are selected for their desirable agricultural attributes [16] is limited by their low rhizogenic capacity, such as the peach rootstock ‘Lovell’ and *Prunus mume* rootstocks [24]. We conducted this study to compare the ability of AR formation in different peach-based hybrid rootstocks (four *P. dulcis* × *P. persica* and one *P. davidiana* × *P. persica*) to contribute to the understanding of the physiological mechanisms and environmental signals that regulate this process, which will allow the identification of tools aimed at improving the vegetative propagation of elite rootstocks.

In the commercial substrate, hardwood cuttings from Adafuel and Garnem showed a highly developed AR system compared to the other three genotypes after treatment with exogenous auxins. Among the genotypes studied, Adarcias and Cadaman were not able to produce any regenerative response after 82 days on the commercial substrate. The genetic background of Cadaman and the morphological characteristics of Adarcias, which are known to be closer to peach, could explain their poor rooting performance. These results confirmed that the rooting performance of hardwood cuttings on *Prunus* rootstocks is highly dependent on their genotype [25].

Next, we established a hydroponic system that allowed us to quantify several parameters related to rooting performance in hardwood cuttings at different times (32, 50, and 90 dap). Callus formation, callus size, and AR number allowed us to classify the studied rootstocks according to their regenerative response. At the end of the experiment, Adafuel showed the highest proportion of callus formation, followed by Adarcias, Garnem, and GF 677, all three with similar levels. Adafuel and Garnem showed a higher percentage of rooting than Adarcias and GF 677, and, in all cases, ARs developed from callus tissue. Cadaman showed the lowest percentage of callus formation and AR formation. This is consistent with the idea that some cells within the callus acquire root identity and act as the founder cells for new AR primordia. Although it is well known that ARs initiate directly from the callus tissue in hardwood cuttings of various species, such as *Eucalyptus* sp. [26] or *Vitis vinifera* [27], we found no correlation between callus size and AR number, suggesting that these two processes (callus growth and establishment of AR primordia) could be regulated independently in *Prunus* rootstocks. Indeed, studies conducted on grape rootstocks also confirmed the independent regulation of callus and AR formation in this species [28]. The rooting of hardwood cuttings from Garnem and GF 677 differed mainly in the timing of AR formation, resulting in a significantly higher number of ARs in Garnem. In addition, we found striking differences in the AR system of hydroponically grown Adafuel, Garnem, and GF 677 hardwood cuttings, particularly with respect to AR number, AR length, and AR growth angle. Adafuel developed longer, thinner ARs with steeper root growth angles than Garnem and GF 677, a root ideotype that provides superior performance under water and nutrient limitations [29]. Furthermore, the regenerative response of Adarcias was limited to callus formation, suggesting that AR initiation is defective in this genotype. In fact, Adafuel was the first peach–almond hybrid released in Spain due to its best rooting ability for clonal propagation [19]. Adarcias was later released due to the control of tree vigor and the better productivity of budded peach cultivars [15]. These two rootstocks have an open-pollinated origin, and further comparative studies will allow the molecular signatures (hormones, metabolites, genes, etc.) of their differential regenerative responses to be identified. In addition, our results from substrate and hydroponic system experiments confirmed that adventitious rooting of hardwood cuttings was dependent on both genotype and environment factors [22,30,31], and as such, these two factors must be considered during rootstock breeding.

To characterize the physiological and molecular events that occur during the rooting of stem cuttings from contrasting *Prunus* rootstocks, we selected Garnem and GF 677 for detailed studies using environmentally controlled in vitro conditions. We found that cutting the basal region of shoot explants could be used as a landmark to trigger new regenerative responses near the wound, regardless of the age of the mother plant or the duration of in vitro incubation. At the ultrastructural level, we found that the internal arrangement of the different tissues near the wound was similar in Garnem and GF 677 microcuttings and that the ARs were formed de novo from some vascular cells located on the external side of the cambium [32], as described in other woody species [33]. Without an exogenous auxin supply, Garnem produced calli at a higher rate than GF 677, resulting in a lower regenerative response at the end of the experiment (30 dap) in GF 677. In both cases, AR emergence occurred several days after callus formation, but the proportion of rooted microcuttings was higher in Garnem. Callus and subsequent AR formation in stem cuttings were dependent on the accumulation of auxin near the wound, which is mainly transported from the shoot [3,34,35]. Therefore, the differences observed between Garnem and GF 677 could be due to differences in auxin levels and/or auxin responses at the base of the stem cuttings. As we found similar levels of active auxin (IAA) in the basal region of microcuttings and during rooting in Garnem and GF 677, the observed differences in regeneration and rooting performance between these two genotypes could be due to differential auxin responses. In fact, exogenous auxin (IBA) treatment of microcuttings enhanced the regenerative response in Garnem to a much greater degree than in GF 677. Treatment with exogenous IBA improved the rooting of GF 677 to the same extent as Garnem without IBA, suggesting that GF 677 is less sensitive to exogenous auxin than Garnem. In addition, endogenous IAA levels were higher in IBA-treated Garnem microcuttings than in IBA-treated GF 677 microcuttings during rooting. To rule out that the differential rooting responses between Garnem and GF 677 could be caused by altered hormonal homeostasis, we measured the endogenous levels of the main plant hormones. Interestingly, the overall levels of some stress hormones, such as ACC (the ethylene precursor), ABA, MeJA, and SA, remained lower during rooting in Garnem compared to those in GF 677. In addition, the levels of GA3 were significantly higher in Garnem than in GF 677. A recent transcriptomic study on AR formation in apple rootstocks suggested that higher ABA and MeJA accumulation and lower GA3 levels could lead to AR inhibition [36]. Based on these results, it is tempting to speculate that a conserved hormonal balance could regulate AR formation in woody species and, as such, could be used as a bioindicator of rooting performance during rootstock breeding.

Although auxins are traditionally used as the main trigger for AR formation in many species [37], the continuous application of IBA had a negative effect on the development of a functional AR system in *Prunus* rootstocks, as it was found that it inhibited AR elongation, as previously described in the microcuttings of apple [38] and wild cherry [39]. This confirms that the exogenous application of auxin helps to extend the induction but can inhibit the growth and development of the AR primordium [33]. IBA is considered a storage form of auxin that is metabolized into free IAA in peroxisomes, and this pathway is elucidated by forward genetic analysis in *Arabidopsis thaliana* [40]. We found that a 24 h-pulse of IBA incubation was sufficient to induce effective rooting in Garnem and GF 677. In fact, endogenous IAA levels were highest at 4 and 8 dae in microcuttings treated with the IBA pulse, suggesting that IBA could be first internalized and then transformed into IAA. Transcriptomic studies in several species suggested that exogenous IBA induced IAA transport through regulation of gene expression [41,42]. Large-scale transcriptome sequencing of the roots of two different *Prunus* rootstocks was performed to identify candidate genes involved in response to root hypoxia [43]. We plan to follow a similar approach to identify the molecular determinants of the contrasting responses observed for Garnem and GF 677 microcuttings during adventitious rooting.

## 4. Materials and Methods

### 4.1. Plant Material, Growth Conditions, and Sample Collection

The plant material used in the hydroponic culture was provided by the Pomology Department of Aula Dei Experimental Station (Zaragoza, Spain). Hardwood cuttings of the following five rootstock genotypes were analyzed: Adafuel (n = 85), Adarcias (n = 72), Cadaman (n = 59), Garnem (n = 67), and GF 677 (n = 84) (Table 1). Hardwood cuttings were taken in the morning (between 9:00 and 10:00 A.M.) on 20 November 2020. They were cut from one-year-old stems sampled from 10-year-old mother plants grown in experimental plots under field conditions. Mother plants were initially established from hardwood cuttings previously rooted and grown in nursery conditions. The thickness of the stem cuttings in each genotype was 7.2 ± 0.3 mm (Adafuel), 7.5 ± 0.2 mm (Adarcias), 8.0 ± 0.3 mm (Cadaman), 7.2 ± 0.3 mm (Garnem), and 6.1 ± 0.3 mm (GF 677). The average length of the stem cuttings in each genotype was 25.9 ± 0.1 cm (Adafuel), 26.3 ± 0.2 cm (Adarcias), 25.9 ± 0.3 cm (Cadaman), 27.0 ± 0.2 cm (Garnem), and 24.7 ± 0.2 cm (GF 677).

To carry out the in vitro assays, microcuttings belonging to Garnem (n = 336) and GF 677 (n = 360) genotypes were provided by Invisa Biotecnología Vegetal S.L. (Caravaca de la Cruz, Murcia, Spain). GF677 and Garnem mother plants were grown in 17 liter pots filled with coconut fiber and peat substrate under the environmental conditions of the rooting station of Invisa Biotecnología Vegetal S.L. at 38°2′13.2″ N, 1°57′35.9″ W and 630 m altitude. Water, fertilizers, and the necessary phytosanitary treatments were applied periodically. In vitro culture explants were established in Spring 2020 according to company standards. Stem cuttings (average length of 5.0 ± 0.9 cm) were collected before 10:00 A.M. by qualified personnel in a laminar flow hood. Plant material was provided in plastic containers with 100 mL of preincubation medium. The preincubation medium contained 5.3 g/L agar, 20 mg/L sucrose, 2 mL/L vitamins, 30 mL/L macronutrients, and 1 mL/L micronutrients; supplemented with iron chelate (Fe EDDHA) (200 mg/L), IBA (0.9 mg/L), inositol (500 mg/L), anthranilic acid (10 mg/L), zinc acetate (10 mg/L), and charcoal (50 mg/L). These microcuttings remained in the preincubation medium from 4 to 22 days before starting the experiments.

#### 4.1.1. Hydroponic Culture of Hardwood Cuttings

For AR induction in the hydroponic culture assay, about 2 cm of the basal region of each hardwood cutting was removed by a bevel cut to renew the tissue. The wounded region was then impregnated with a commercial mix of powdered rooting hormones (0.4% *w/w* IBA, 0.4% *w*/*w* 1-naphthaleneacetic acid (NAA), and the fungicide Captan (15% *w/w*)). Hormone-treated hardwood cuttings were inserted into predrilled expanded polystyrene supports and placed in opaque square containers (18.5 × 18.5 × 13 cm) filled with tap water. Continuous aeration was implemented in each container by connecting an air pump to a rubber tube, which was directly introduced into the water. The hydroponic system was kept in a walk-in growth room under controlled conditions, with a photoperiod of 16 h light (day)/8 h darkness (night) until the end of the experiment. The average temperature was 23.7 ± 2.2 °C (day) and 19.9 ± 1.3 °C (night); and the average relative humidity was 59.0 ± 2.4% (day) and 61.5 ± 1.8% (night). Water was renewed every week. Harwood cuttings (n = 367) were grown in these conditions for 90 days, from 19 November 2020 to 17 February 2021.

#### 4.1.2. In Vitro Culture of Garnem and GF 677 Microcuttings

For the in vitro AR induction assays, about 2 cm of the most basal region of the microcuttings in the preincubation medium was excised using a scalpel and sterile tweezers in a Telstar AH100 horizontal laminar flow cabinet and the apical microcuttings (2.45 ± 0.4 cm length) were transferred to SteriVent containers (Duchefa Biochemie, 10.7 × 9.4 × 9.6 cm) with 100 mL of standard growing medium, as previously described [44]. This incubation medium was supplemented with 0.9 mg/L indole-3-butyric acid (IBA; Duchefa Biochemie) (auxin treatment) or with 0.1 mL absolute ethanol (mock treatment). Twelve stem microcuttings were regularly spaced in each container. The containers were sealed with 3 M surgical Micropore tape and kept in a growth cabinet with an established photoperiod of 16 h light/8 h darkness. The average temperature was 23.7 ± 2.5 °C (day) and 19.9 ± 1.8 °C (night); and the average relative humidity was 56.7 ± 4.0% (day) and 59.7 ± 2.9% (night).

A total of 696 microcuttings were used for the in vitro culture assays, which were divided into three experimental replicates: n = 144 (E1), n = 360 (E2), and n = 192 (E3). In E1 and E2, the continuous application of IBA was compared to the mock treatment, while in E3, the application of a 24 h pulse of IBA (IBAp) was compared to the mock treatment. Microcuttings in E1 were studied between 15 December 2020 to 14 January 2021; microcuttings in E2 were studied between 10 March to 30 March 2021; and microcuttings in E3 were studied between 19 April to 9 May 2021 (Appendix A).

#### 4.1.3. Direct Rooting on Substrate

Three 13 cm diameter pots with four microcuttings each and containing 100 g of 1:1 (*v*/*v*) peat/perlite substrate and 1.5 g of Osmocote per genotype and treatment were maintained for 25 days in the growth cabinet under controlled conditions of 16 h light (average photosynthetic photon flux density of 50 m^−2^ s) at 26.0 ± 0.9 °C and 8 h darkness at 23 ± 1.0 °C.

#### 4.1.4. Sample Collection

For hormone analysis in the in vitro assay, three replicates of each genotype, treatment, and time point were taken, each consisting of 5 mm-length basal sections of 6 microcuttings. Samples were collected at excision time (T0), 4 days after excision in the incubation medium (T4), and 8 days after excision (T8). Samples were immediately frozen in liquid nitrogen and stored at −80 °C.

### 4.2. Histology

For histological analysis, serial sections from the most basal region (5 mm) of several microcuttings of Garnem and GF 677 were obtained using a razor blade. These sections were fixed in paraformaldehyde solution (1.85% *v/v* formaldehyde, 45% ethanol, 5% acetic acid, and 1% Triton X-100) overnight at 4 °C. Fixed tissues were rinsed three times in 0.1 M Sorensen’s phosphate buffer for 5 min. Samples were then embedded in ClearSee solution [45] and stored cold (4 °C) overnight. The samples were immersed in 5 mL of 0.1% calcofluor white and incubated for 2 h at room temperature. The sections were washed five times using ClearSee solution (10 min each). The samples were observed in the following days using a Motic BA210 brightfield microscope (Motic Spain, Spain), and images were captured with an integrated Moticam 580INT documentation station (Motic Spain).

### 4.3. Picture, Image Processing, and Parameter Measurement

Pictures of hardwood cuttings from hydroponic culture were taken at 32 days after planting (dap) using an Epson Perfection V330 Photo scanner at a resolution of 720 ppi (pixels per inch). Images at 50 and 90 dap were obtained with a Sony Cyber-shot DSC-H 8.1 Megapixel digital camera.

For the in vitro assays, images of the basal part of the microcuttings were taken in the incubation medium at 0, 7, 8, 9, 13, 15, 20, and 30 days after excision (dae) (Appendix A) using the Epson Perfection V330 Photo scanner at a resolution of 600 ppi (pixels per inch). For experimental replicates E2 and E3, the remaining microcuttings that were not used for sample collection were removed from the incubation medium and photographed with the Sony Cyber-shot DSC-H 8.1 Megapixel digital camera (Appendix A).

Most regeneration traits were visually quantified from the stored images (e.g., presence of callus, callus stage, root number) or calculated from previous parameters (e.g., rooting percentage or callus percentage). Image processing with GIMP (GIMP Development Team, 2019) and Fiji ImageJ [46] allowed us to measure other quantitative traits such as callus area and maximum AR length. Studied traits in hydroponic and in vitro experiments are shown in Appendix A, respectively.

### 4.4. Phytohormone Extraction and Analysis

The main active plant hormones and some hormone derivatives and conjugates were analyzed as previously described [47,48] with some modifications. Cold stored samples were freeze-dried in liquid nitrogen and ground with a pestle into a coarse powder. Powdered samples were homogenized in 1.5 mL of cold (−20 °C) extraction mixture of methanol/water (80/20, *v/v*). Solids were separated by centrifugation (20,000× *g*, 15 min) and re-extracted for 30 min at 4 °C in an additional 1.5 mL of the same extraction solution. Pooled supernatants were passed through Sep-Pak C18 Plus short cartridges (SepPak Plus, Waters Corporation, Milford, MA, USA) to remove interfering lipids and part of the plant pigment. They were evaporated at 40 °C under vacuum either to near dryness or until the organic solvent was removed. The residue was dissolved in 0.2 mL of methanol/water (20/80, *v/v*) solution using an ultrasonic bath. The dissolved samples were filtered through 13 mm diameter Millex filters with nylon membrane (0.22 µm pore size, Millipore, Bedford, MA, USA).

Ten microliters of filtrated extract were injected into a U-HPLC-MS system consisting of an Accela Series U-HPLC (ThermoFisher Scientific, Waltham, MA, USA) coupled to an Exactive mass spectrometer (ThermoFisher Scientific, Waltham, MA, USA) using a heated electrospray ionization (HESI) interface. Mass spectra were obtained using Xcalibur software, version 2.2 (ThermoFisher Scientific, Waltham, MA, USA). For the quantification of the plant hormones, calibration curves were constructed for each analyzed component (1, 10, 50, and 100 µg L^−1^) and corrected for 10 µg L^−1^ deuterated internal standards. Recovery percentages ranged between 92% and 95%. Hormone derivatives and conjugates were identified by extracting the exact mass from the full scan chromatogram obtained in negative mode and adjusting a mass tolerance of ≤1 ppm. The concentrations were semi-quantitatively determined from the extracted peaks using calibration curves of analogue hormones. The raw data for phytohormone extraction is shown in Appendix A.

### 4.5. Statistical Analyses

Statistical analyses of data and descriptors (mean, standard error of the mean [SEM], maximum and minimum, and correlation values) were estimated with StatGraphics Centurion XVI version 16.1.03 (StatPoint Technologies, Warrenton, VA, USA). Data outliers were identified based on aberrant standard deviation values and were excluded for posterior analyses, as described elsewhere [49]. To compare the data for a given variable, we performed multiple testing analyses with the ANOVA F-test or the Fisher’s Least Significant Difference (LSD) post hoc test, as indicated. Significant differences were defined as a 1% level of significance (*p*-value < 0.01) unless otherwise indicated. To establish the correlation between different parameters, multiple correlation tests were carried out for the selected parameters.

## 5. Conclusions

The application of exogenous auxin led to an improvement in the rooting performance of the genotypes studied. While continuous IBA exposure promoted callus formation, a 24 h IBA pulse (IBAp) promoted highly efficient AR formation in *Prunus* microcuttings. IBAp treatment was easily applied, and the effect was proven to be reproducible in our experiments; therefore, it is highly recommended for commercial in vitro propagation strategies. An integrative approach based on selective transcriptome profiling will allow us to identify the key regulatory pathways involved in AR performance in *Prunus* species, which could then be used to breed rootstocks with enhanced rooting ability to increase production under challenging environmental conditions.

## Figures and Tables

**Figure 1 plants-11-00913-f001:**
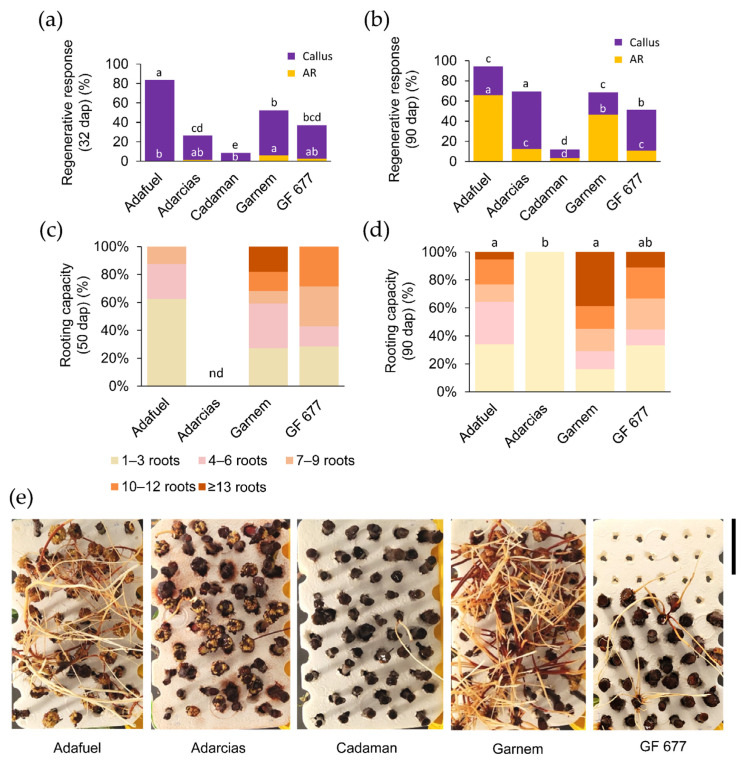
Rooting performance of hardwood cuttings of five *Prunus* rootstocks during hydroponic growth. (**a**,**b**) The regenerative response was estimated by the percentage of cuttings with callus and adventitious roots (ARs) at (**a**) 32 days after planting (dap) and (**b**) 90 dap. The number of samples varies from 59 to 72. Letters indicate significant differences (*p*-value < 0.05; Chi-square) between genotypes in callus regenerative response, and white letters indicate significant differences (*p*-value < 0.05; Chi-square) between genotypes in AR regenerative response. (**c**,**d**) Rooting capacity at (**c**) 50 and (**d**) 90 dap according to the indicated categories. Letters indicate significant differences (*p*-value < 0.01) for the average number of ARs between the genotypes studied. The number of samples varies from 7 to 56. (**e**) Representative images of the genotypes studied after 90 days of growth in hydroponic culture. Scale bar: 7.5 cm.

**Figure 2 plants-11-00913-f002:**
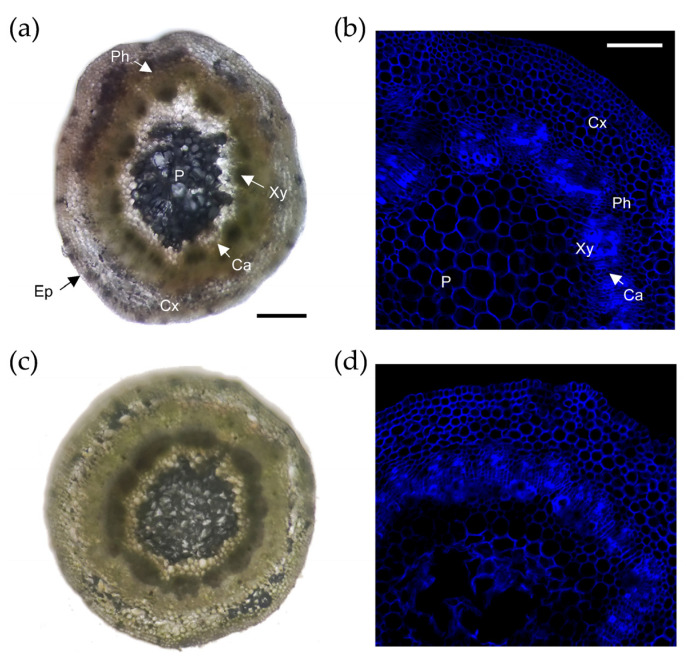
Histological comparison between Garnem and GF 677 microcuttings. No structural differences were observed between Garnem (**a**,**b**) and GF 677 (**c**,**d**) tissue microcuttings before AR formation. Scale bar 200 µM (**a**,**c**) and 100 µM (**b**,**d**). Cx, cortex; Ph, phloem; Ca, cambium; Xy, xylem; P, pith; and Ep, epidermis.

**Figure 3 plants-11-00913-f003:**
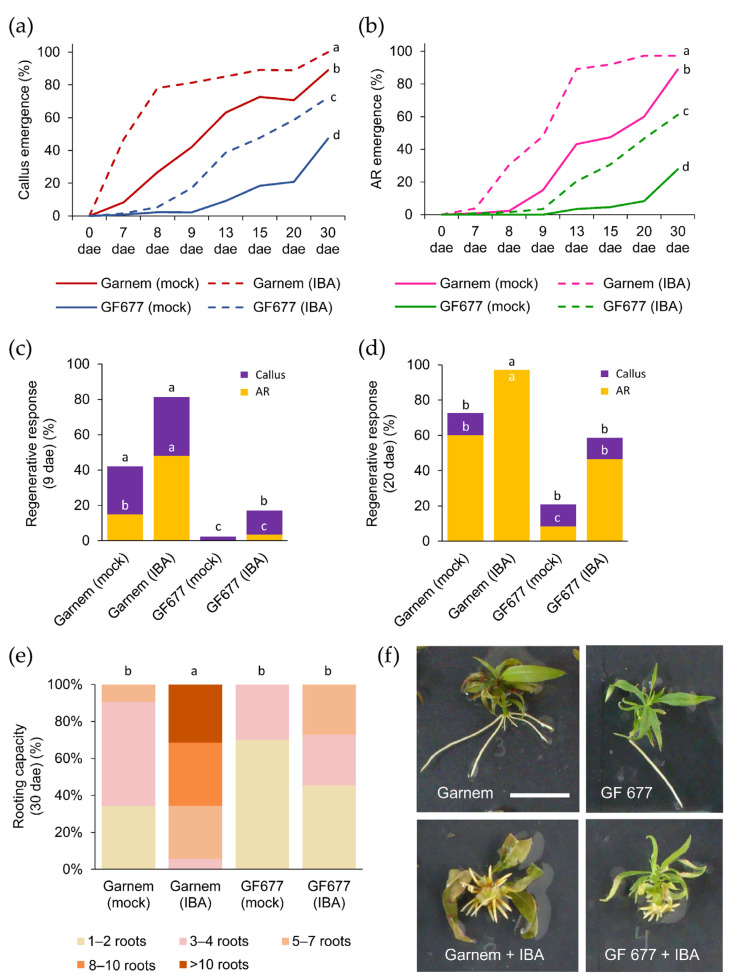
Rooting performance of Garnem and GF 677 microcuttings grown in vitro. (**a**,**b**) Appearance of (**a**) callus and (**b**) ARs over time in microcuttings treated with and without auxin (0.9 mg/L IBA). Letters indicate significant differences (*p*-value < 0.05) at 30 days after excision (dae). (**c**,**d**) Regenerative response estimated by the percentage of cuttings with callus and ARs at (**c**) 9 days after excision (dae) and (**d**) 20 dae. The number of samples varies from 36 to 100. Letters indicate significant differences (*p*-value < 0.05; Chi-square) between genotypes in callus regenerative response, and white letters indicate significant differences (*p*-value < 0.05; Chi-square) between genotypes in AR regenerative response. (**e**) Rooting capacity measured at 30 dae. Letters indicate significant differences (*p*-value < 0.01) for the average number of ARs. (**f**) Representative images of Garnem and GF 677 microcuttings at the end of the experiment. Scale bar: 2.5 cm.

**Figure 4 plants-11-00913-f004:**
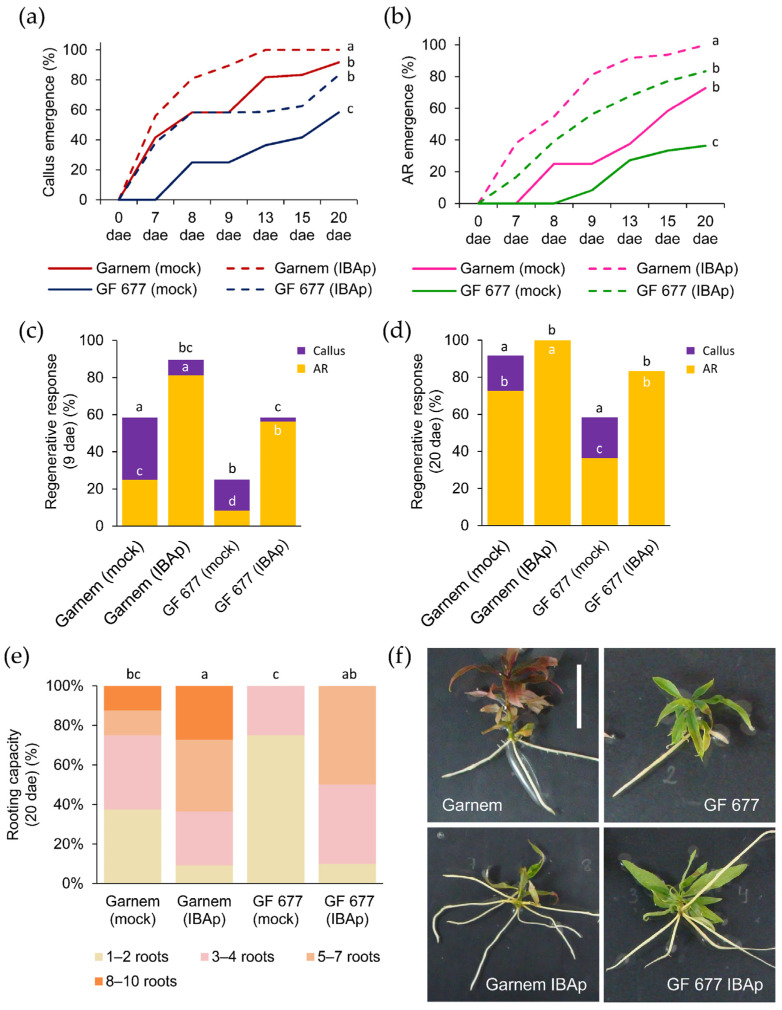
Rooting performance of Garnem and GF 677 microcuttings after a 24 h pulse of IBA. (**a**,**b**) Appearance of (**a**) callus and (**b**) ARs over time in microcuttings treated with and without a 24 h pulse of IBA 0.9 mg/L (IBAp). Letters indicate significant differences (*p*-value < 0.05) between categories. (**c**,**d**) Regenerative response estimated by the percentage of cuttings with callus and ARs at (**c**) 9 days after excision (dae) and (**d**) 20 dae. The number of samples varies from 11 to 48. Letters indicate significant differences (*p*-value < 0.05; Chi-square) between genotypes in callus regenerative response, and white letters indicate significant differences (*p*-value < 0.05; Chi-square) between genotypes in AR regenerative response. (**e**) Rooting capacity measured at 20 dae. Letters indicate significant differences (*p*-value < 0.01) for the average number of ARs. The number of samples in the rooting capacity graph varies from 11 to 12. (**f**) Representative images of Garnem and GF 677 microcuttings at the end of the experiment. Scale bar: 2.5 cm.

**Figure 5 plants-11-00913-f005:**
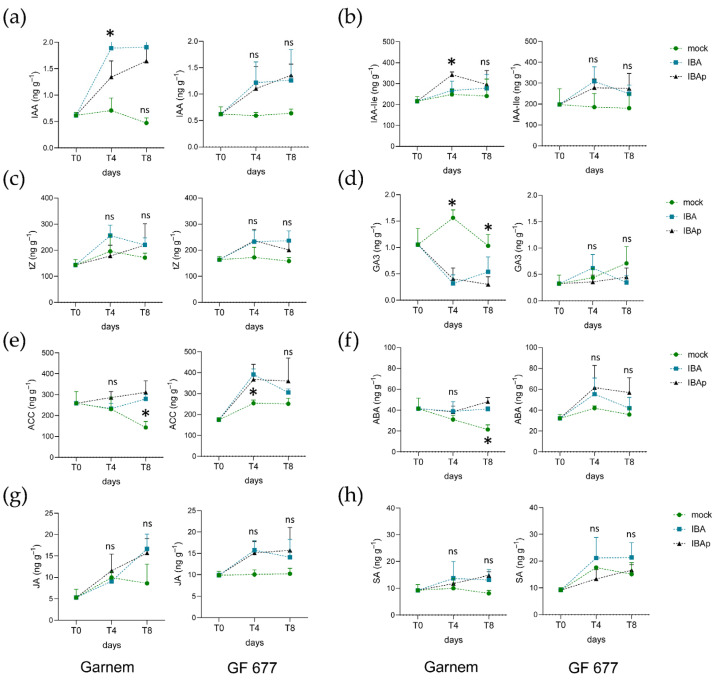
Hormonal profile of Garnem and GF 677 microcuttings during IBA-induced rooting. Endogenous levels of (**a**) indole-3-acetic acid (IAA), (**b**) IAA-Isoleucine (IAA-Ile), (**c**) trans-zeatin (*t*Z), (**d**) gibberellic acid 3 (GA3), (**e**) 1-aminocyclopropane 1-carboxylic acid (ACC), (**f**) abscisic acid (ABA), (**g**) jasmonic acid (JA), and (**h**) salicylic acid (SA) measured in the basal region of Garnem (left panels) and GF 677 (right panels) microcuttings at 0, 4, and 8 dae. Asterisks indicate statistically significant differences (*p*-value < 0.01) between treatments at the same time point, ns = no statistically significant differences (*p*-value < 0.01).

**Table 1 plants-11-00913-t001:** *Prunus* rootstocks used in this work.

Rootstock Name	Species	Origin
Adafuel	*P. dulcis x P. persica*	CSIC, Spain
Adarcias	*P. dulcis x P. persica*	CSIC, Spain
Cadaman	*P. persica x P. davidiana*	IFGO, Hungary and INRA, France
Garnem	*P. dulcis x P. persica*	CITA, Spain
GF 677	*P. dulcis x P. persica*	INRA, France

CSIC, Consejo Superior de Investigaciones Científicas. INRA, Institut National de la Recherche Agronomique. CITA: Centro de Investigación y Tecnología Agroalimentaria de Aragón.

**Table 2 plants-11-00913-t002:** RR50 values (expressed in dae) obtained for callus and AR emergence from E1 and E2.

Genotype (Treatment)	Callus Emergence	AR Emergence
Garnem (mock)	9.7	15.7
Garnem (IBA)	6.3	8.9
GF 677 (mock)	>30	>30
GF 677 (IBA)	12.7	16.3

**Table 3 plants-11-00913-t003:** RR50 values (expressed in dae) obtained for callus and AR emergence from E3.

Genotype (Treatment)	Callus Emergence	AR Emergence
Garnem (mock)	7.6	13.5
Garnem (IBAp)	6.7	7.5
GF 677 (mock)	>20	>20
GF 677 (IBAp)	11.9	8.1

## Data Availability

Data supporting reported results can be found in the Appendix A.

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
