# Peer review of "Effects of Auxin (Indole-3-butyric Acid) on Adventitious Root Formation in Peach-Based Prunus Rootstocks"

_plants, 2022, doi:10.3390/plants11070913_

Round 1

Reviewer 1 Report

I have revised the article entitled “Characterization of Adventitious Root Formation in Prunus Rootstocks”. Authors assessed the adventitious root development in different Prunus species and the auxin-induced callus formation in the same species. The manuscript is very interesting; however, little concern could be revised:

Title

The title resulted very generic, and a more specific sentence could be more appropriate and useful in the future citation of the present manuscript.

Introduction

Line 76: [9] specify authors in the introduction: Felipe et al. [9]

Lines 93-96: this sentence is more appropriate in Conclusions section.

Materials and Methods

Paragraph 4.1: More explanation of experimental design is needed (how many plants were used, etc.).

Lines 504-517: Specify in which the hydroponic culture consisted of.

Line 612: which kind of ANOVA? Specify and add “post-hoc LSD method”. Specify also which source of variability was used in the ANOVA analysis.

I suggest the acceptation of the present manuscript with minor modifications.

Author Response

See attached document for detailed point-by-point responses.

Reviewer 2 Report

The manuscript titled „Characterization of Adventitious Root Formation in Prunus Rootstocks” is about the rooting of Prunus rootstocks during the propagation. This step is very difficult to „solve” in the practice, therefore this step determinates the effectiveness and usage of the propagation methods of the Prunus rootstocks.

In the Table1 there is a little mistake. The correct origin of Cadaman is INRA France – Hungary, the GF 677 is bred at INRA France. Please double check the “Species” category of the Table 1, because there are some mistakes in there.

How were the mother plants propagated (in vitro or cuttings), from which the plant materials were collected? When did you collect the hardwood and the microcuttings?

The discussion contains a lot of useful information about the relationship between the results and the genetic background of the rootstocks. These connections can be very useful in the future too.

Author Response

(The authors gave the same response as above.)
